# STAIR: Learning Sparse Text and Image Representation in Grounded Tokens

**Chen Chen,  Bowen Zhang,  Liangliang Cao,  Jiguang Shen,**
**Tom Gunter,  Albin Madappally Jose,  Alexander Toshev,  Yantao Zheng,**
**Jonathon Shlens[†],  Ruoming Pang,  Yinfei Yang**
Apple AI/ML

{chen_chen999, bowen_zhang4, lcao27, yinfeiy}@apple.com

## Abstract

Image and text retrieval is one of the foundational tasks in the vision and language domain with multiple real-world applications. State-of-the-art contrastive approaches, *e.g.* CLIP (Radford et al., 2021), ALIGN (Jia et al., 2021), represent images and texts as dense embeddings and calculate the similarity in the dense embedding space as the matching score. On the other hand, sparse semantic features like bag-of-words models are inherently more interpretable, but believed to suffer from inferior accuracy than dense representations. In this work, we show that it is possible to build a sparse semantic representation that is as powerful as, or even better than, dense presentations. We extend the CLIP model and build a sparse text and image representation (STAIR), where the image and text are mapped to a sparse token space. Each token in the space is a (sub-)word in the vocabulary, which is not only interpretable but also easy to integrate with existing information retrieval systems. STAIR model significantly outperforms a CLIP model with +4.9% and +4.3% absolute Recall@1 improvement on COCO-5k text→image and image→text retrieval respectively. It also achieved better performance on both of ImageNet zero-shot and linear probing compared to CLIP. [1]

## 1 Introduction

Learning high-quality and performant representations from large-scale image-text data has been intensively studied in recent years. Improved vision-language representation benefits many applications, including image-text retrieval (Chen et al., 2015; Plummer et al., 2015), VQA (Antol et al., 2015; Johnson et al., 2017), and image captioning (Vinyals et al., 2015). The resurgence of interest in contrastive learning has fueled recent advances. State-of-the-art models like CLIP (Radford et al., 2021) and ALIGN (Jia et al., 2021) employ

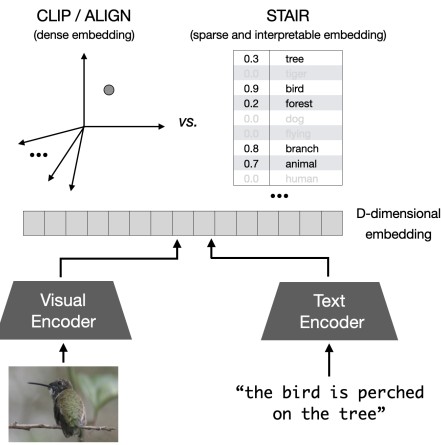

Figure 1: **Learning a sparse and interpretable embedding for a vision-language model.** CLIP (Radford et al., 2021) and ALIGN (Jia et al., 2021) learn a compact dense embedding, while STAIR aims to learn a sparse and interpretable embedding in a high-dimensional space. Each dimension in the sparse embedding represents a (sub-)word from a large dictionary, with a non-negative importance weight.

dedicated encoders for images and text, enabling joint embedding in a shared space. By enhancing the cosine similarity for aligned pairs and dissimilarity for unmatched pairs, these contrastive models achieve remarkable performance in fine-tuning and zero-shot generalization for image-text retrieval, VQA, and image classification tasks.

Despite the impressive benchmark performance, the dense embedding space is usually considered as a black box and challenging to interpret. The meaning of an dense embedding is determined by its vector relationships with other embeddings, rather than corresponding directly to any human-understandable concept. This lack of direct correspondence with interpretable concepts hinders the transparency and interpretability. Moreover, deploying a retrieval model trained with a contrastive objective to real-world large-scale image-text retrieval is a non-trivial task. Despite approximated

---

[1] [†]Work done while at Apple

nearest neighbor search (Guo et al., 2020; Johnson et al., 2019a) can be used to retrieve from a dense embedding space, the cost can be high when scaling up to billions of images. Additionally, traditional approaches like inverted indexes, commonly employed by search engines, cannot be straightforwardly applied to the dense embeddings. The lack of interoperability further complicates the integration of dense embeddings with other retrieval features without additional training. In comparison, interpretable sparse embeddings possess distinct advantages across numerous applications. More detailed discussion can be found in Appendix D.

The community has explored sparse and interpretable image representations extensively. Early approaches, *e.g.* bag-of-words and topic models (Csurka et al., 2004; Fei-Fei and Perona, 2005; Lazebnik et al., 2006) were widely coupled with SIFT descriptor (Lowe, 2004) but found to exhibit inferior performance compared to dense vectors (Lin et al., 2011; Sánchez et al., 2013). Other endeavors, like deep visual-semantic embedding (Frome et al., 2013) using ImageNet topics yielded more interpretability but still failed to outperform dense representations (Faghri et al., 2017).

We hypothesize that the gap between sparse semantic and dense embedding stems from two primary factors: (1) Previous works on semantic embedding inadequately explored large-scale training to capture the rich semantics in the image-text domain. (2) Most existing semantic embedding methods are built on a fixed vocabulary (e.g. thousand of concepts), which cannot handle out-of-vocabulary concepts. In this paper, we present a new model, named **STAIR**, and a multi-stage training recipe to learn a **S**parse **T**ext **A**nd **I**mage **R**epresentation to tackle the aforementioned challenges. Our sparse representation not only matches but also outperforms the dense image-text representations.

Inspired by the recent success of sparse representation in information retrieval field (Bai et al., 2020; Formal et al., 2021b), STAIR encodes the image and text into a sparse and grounded token space, as illustrated in Figure 1. Specifically, images and text are mapped to sparse and interpretable in a high-dimensional space, where each dimension is associated with an actual (sub-)word from a large dictionary, with non-zero weights. Notably, the proposed multi-stage training recipe plays a critical role in the grounding of the sparse embedding.

We conduct a comparative analysis between the STAIR model and a CLIP model, sharing the same architecture and dataset. Experiment results a significant improvement of STAIR model over CLIP on image-text retrieval tasks, with +4.9% and +4.3% recall@1 on COCO-5K (Chen et al., 2015) text→image and image→text retrieval respectively. STAIR models also demonstrate similar or better performance on zero-shot classification and linear probing tasks including ImageNet. To quantify the interpertability of the embeddings, we define an interpretable space using BERT vocab. Experiments suggest STAIR is substantially more interpretable, achieving a Top-1 accuracy 32.9% on ImageNet, outperforming CLIP's accuracy of 13.7%.

## 2 Approach

We begin with an overview of the dual-encoder architecture to establish the basis of our research.

### 2.1 Dual-Encoder Contrastive Learning

Given a dataset of image-text pairs $D = \{(x_i, y_i)\}$, where $x_i$ and $y_i$ represent the image and text respectively, a dual encoder learns a similarity $M(x, y)$, such that the aligned pair $(x_i, y_i)$ is assigned higher similarity score compared to the unmatched pairs sampled from a negative sample set $D_i'$.

A dual-encoder architecture comprises an image encoder $E_{\text{IMAGE}}$ and a text encoder $E_{\text{TEXT}}$. Each encoder $E$ consists of two components: 1) a standard input-specific neural network $f(\cdot)$ and 2) a projection head $g(\cdot)$ that maps the features into embeddings in a joint dense embedding space:

$$E(\cdot) = g(f(\cdot)) \tag{1}$$

where $f(\cdot)$ is typically a transformer model and $g(\cdot)$ is a pooling layer.

The similarity between an image $x$ and text $y$ is measured by cosine similarity:

$$M(x, y) = \frac{E(x)^T E(y)}{\|E(x)\|\|E(y)\|} \tag{2}$$

The dual encoder is trained by minimizing the contrastive loss $\mathcal{L}_{\text{CON}}$:

$$\mathcal{L}_{\text{CON}} = -\frac{1}{|D|} \sum_i \log \frac{e^{M(x_i, y_i)/T}}{\sum_{(x', y') \in D_i} e^{M(x', y')/T}} \tag{3}$$

where $D_i = D_i' \cup \{(x_i, y_i)\}$ denotes the positive pair and its negative set for each pair, and $T$ is the softmax temperature.

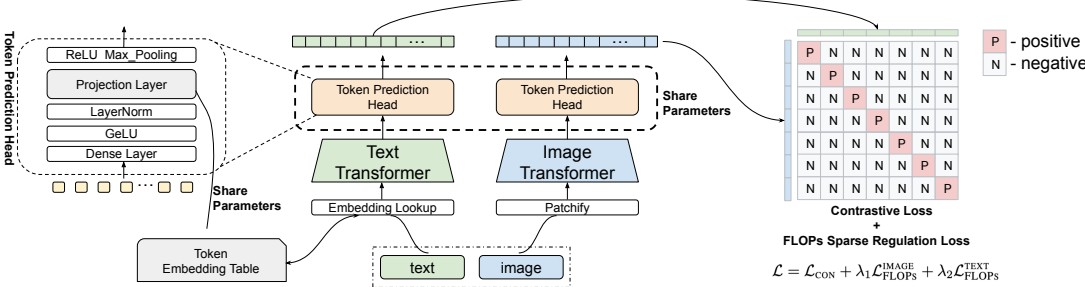

Figure 2: **Diagram of STAIR architecture.** It employs a dual-encoder architecture. Different than dense models like CLIP or ALIGN, STAIR maps the dense embeddings to a sparse latent embedding space via a token prediction head. In addition to a regular contrastive loss to minimize an image-text matching loss, a FLOPs (Paria et al., 2020) loss is added to encourage sparsity of image and text embeddings.

## 2.2 STAIR

Following CLIP (Radford et al., 2021), STAIR also adopts a dual-encoder architecture. As depicted in Figure 2, it consists of an image encoder and a text encoder, both comprising a feature extraction network $f(\cdot)$ and a projection head $g(\cdot)$. In particular, the dense project head $g(\cdot)$ is replaced with a **Token Projection Head**, which maps the representation to a sparse embedding space. A vocabulary $V$ is used as the basis of embedding space for interpretability purpose.

The token projection head $g(\cdot)$ comprises two components: (1) a mapping function that maps the input sequence $h$ to a sequence of weights for each token $j$ in the vocabulary space $V$, and (2) a pooling layer that summarizes the sequence into a sparse embedding in the vocabulary space $V$.

For the mapping function, we leverage the BERT (Devlin et al., 2019) masked language model (MLM) prediction head $p(\cdot)$:

$$p(h_j) = e \cdot \text{TRANSFORM}(h_j) + b \qquad (4)$$

where $h_j = f(\cdot)$ corresponds to the $j^{\text{th}}$ token in the sequence of the feature extraction network output. The TRANSFORM function comprises a FC layer with GELU activation and a LAYER NORM layer, and $e$ and $b$ are the linear mapping and bias in MLM prediction head that maps the output to vocabulary space. The weights of $e$ is tied with token embedding lookup table in the text encoder.

For the pooling layer, we adopt the approach from Formal et al., 2021b,a to aggregate the weight of token $j$ to sparse embedding ENC:

$$\text{ENC} = \log(1 + \text{RELU}(\max_j(p(h_j)))) \qquad (5)$$

RELU activation non-negativity of token weights

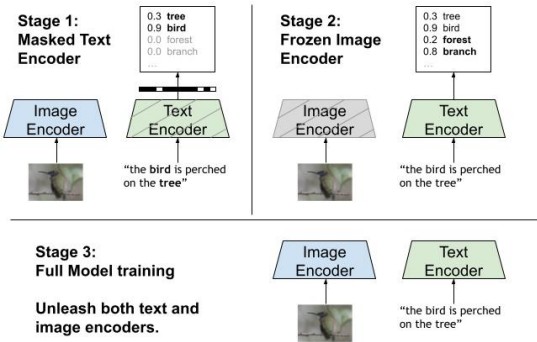

Figure 3: **Training strategy for STAIR model.** (1) The text output is masked to only predict the weights of tokens that occur in the input. (2) The image encoder is frozen and the mask is removed from the text encoder. (3) All constraints are removed.

and adding the log function empirically improves performance by suppressing overwhelmingly large weights (Zhao et al., 2021). Consequently, the image embedding $\text{ENC}^{\text{IMAGE}}$ and text embedding $\text{ENC}^{\text{TEXT}}$ are sparse vectors residing in a $|V|$-dimensional space defined by the vocabulary.

In practice, relying solely on the ReLU operator in Eq. (5) often leads to insufficient sparsity. To obtain better control over sparsity, we incorporate FLOPs regularization loss (Paria et al., 2020) to encourage a small number of non-zeros token embeddings in $V$:

$$\mathcal{L}_{\text{FLOPs}} = \sum_{k \in V} \left( \frac{1}{N} \sum_{i=1}^{N} \text{ENC}_k^{(i)} \right)^2 \qquad (6)$$

By combining the contrastive loss and the FLOPs loss, the STAIR model is optimized by:

$$\mathcal{L} = \mathcal{L}_{\text{CON}} + \lambda_1 \mathcal{L}_{\text{FLOPs}}^{\text{IMAGE}} + \lambda_2 \mathcal{L}_{\text{FLOPs}}^{\text{TEXT}} \qquad (7)$$

where $\lambda_1$ and $\lambda_2$ are FLOPs regularization weights

Table 1: **Model Size (Unit: Millions).** The number of parameters of STAIR is comparable to CLIP.

| | Text Encoder | Image Encoder |
|---|---|---|
| CLIP | 53.7 | 88.6 |
| STAIR | 53.8 | 86.8 |

for image and text embeddings, respectively. Notably, by adjusting these weights, STAIR embeddings can achieve a high level of sparsity, resulting in an effective embedding size even smaller than CLIP. Details are discussed in Section 6.3.

## 3 Training Details

The STAIR model aims to achieve two goals: 1) aligning text and images in the sparse embedding space; 2) grounding the sparse embedding dimension with human-understandable (sub-)word in the vocabulary. In other words, the image and text embeddings should reside in a space defined by basis vectors that correspond to interpretable tokens. However, in practice, we found that simply replacing the dense projection head with a token prediction head alone does not guarantee the 2nd goal outright. This is because the images and text are distinct modalities with inherent semantic gaps, and contrastive loss alone only encourages text/image alignment. We shown in Section 5.2 that the model learns a shortcut by relying on less common tokens to bridge the modality gap.

To address this challenge, we propose a multi-stage training approach that sequentially bridges the discrepancy between the sparse embedding space and the interpretable space defined by the vocabulary as illustrated in Figure 3.

**Stage 1: Training image embedding with masked tokens** In the first stage, we co-train both encoders while applying a binary mask to the text embedding. Formally, given the text input $y$, the masked text embedding is formulated as:

$$\text{ENC}_{\text{MASK}}^{\text{TEXT}} = g(f(y)) \cdot \text{MASK}(y) \qquad (8)$$

where $\text{MASK}_i = 1$ if the $i^{\text{th}}$ token exists in $y$, and $g(f(y))$ predicts the weights of the non-masked tokens. In other words, the text embedding is forced to activate the only the tokens appearing in the original text input, while disregarding others. By matching with the masked text embedding, the image encoder learns to ground its image embedding on the tokens from the paired text. Consequently, after the stage 1, it anchors the image embeddings

to meaningful tokens in the interpretable space defined by $V$.

**Stage 2: Training with frozen image encoder** In this stage, out focus shifts to grounding the text embedding to the same interpretable space. The key idea is to leverage the image encoder teach the text encoder as a teacher model to guide the training of the text encoder. Specifically, we freeze the image encoder while training the text encoder to align its image embedding using contrastive loss. After stage 2 training, both image and text embeddings are positioned within the same human-interpretable embedding space constructed by $V$.

**Stage 3: Joint fine-tuning of both encoders** The initial two-stage training establishes a solid foundation for both encoders to produce human-interpretable embeddings. In stage 3, we enhance the performance of image-text matching by jointly finetuning both encoders.

Experiments show that this multi-stage training recipe is critical for the embedding interpretability. A qualitative and quantitative comparison between training with multi-stage recipe and without can be found in Section 6.1 and Appendix C.

## 4 Experiments

### 4.1 Datasets

Our dataset is a combination of internal and public datasets, totaling 1.1B image-text pairs. The public datasets consists of Conceptual Caption 3M (Sharma et al., 2018) and Conceptual Captions 12M (Changpinyo et al., 2021). The internal dataset consists of 1B image-text pairs, including a 134M clean licensed dataset (see Appendix A for details) and a 971M noisy web-crawled dataset. The web-crawled dataset is mined using a similar approach as described in ALIGN (Jia et al., 2021). We further filter the data using a public CLIP model[2], removing pairs with a CLIPScore (Hessel et al., 2021) below 0.2.

### 4.2 Configurations

In the experiment, we train a CLIP model and a STAIR model for comparison. For both models, we utilize transformer (Vaswani et al., 2017) as the backbone with modified CLIP-B/16 (Radford et al., 2021) configurations. The text encoder is a 12-layer Transformer with 512 hidden dimensions

---

[2]https://huggingface.co/openai/clip-vit-base-patch16

Table 2: **Zero-shot text/image retrieval.** Reporting recall@K on Flickr30K and COCO.

| | COCO 5K | | | | | | Flickr30K | | | | | |
| | text → image | | | image → text | | | text → image | | | image → text | | |
| | R@1 | R@5 | R@10 | R@1 | R@5 | R@10 | R@1 | R@5 | R@10 | R@1 | R@5 | R@10 |
|---|---|---|---|---|---|---|---|---|---|---|---|---|
| CLIP | 36.2 | 62.2 | 72.2 | 53.4 | 78.3 | 85.6 | 63.0 | 86.7 | 92.5 | 79.6 | 95.5 | 98.1 |
| STAIR | **41.1** | **65.4** | **75.0** | **57.7** | **80.5** | **87.3** | **66.6** | **88.7** | **93.5** | **81.2** | **96.1** | **98.4** |

Table 3: **Zero-shot classification accuracy.** Reporting the top-1 accuracy (%) across 9 datasets.

| | ImageNet | Caltech-101 | CIFAR-100 | SVHN | DTD | OxPet | OxFlowers | Eurosat | RESISC45 |
|---|---|---|---|---|---|---|---|---|---|
| CLIP | 65.1 | 82.3 | 63.2 | 42.0 | 53.6 | 85.8 | 67.7 | **52.4** | **64.3** |
| STAIR | **65.6** | **82.5** | **63.4** | **53.0** | **56.3** | **85.9** | **68.2** | 51.0 | 62.8 |

and 8 attention heads, while the image encoder is a 12-layer ViT with 768 hidden dimensions and 12 attention heads. The text input is tokenized using BERT WordPiece tokenizer (Devlin et al., 2019) with a vocabulary of 30,522 tokens. The maximum text input sequence length is set to 76.

The CLIP model is trained for 600K steps using the LAMB optimizer (You et al., 2020) with a learning rate of $1e^{-3}$ and a weight decay of $1e^{-2}$. The learning rate is warmed up for the first 10k steps and then linear decay to 0. The STAIR model goes through 3 stages, with each stage trained for 300K, 300K, and 600K steps, using the same configuration as CLIP. To mitigate catastrophic forgetting, a smaller max learning rate of $1e^{-4}$ is adopted in stage 3. FLOPs regularization weights are set to $\lambda_1 = \lambda_2 = 1e^{-3}$ by default, following the quadratic growth in Formal et al., 2021a. Unless specified otherwise, all models are trained using a global batch size of 16,384. All our experiments were conducted using 64 A100 GPUs.

It is noted that the proposed STAIR model does not introduce additional parameters compared to CLIP, because the linear projection $e$ for both the image and text modalities parameters with token embedding lookup table as described in Section 2.2. The detailed number of learnable parameters for each model are summarized in Table 1.

### 4.3 Zero-Shot Text Image Retrieval

Table 2 shows the recall@K (K=1, 5, 10) performance of image/text retrieval tasks on Flickr30K (Plummer et al., 2015) and COCO-5K (Chen et al., 2015). The metrics are reported with a prompt of "a photo of " added before the original caption, following Radford et al. (2021). We observe a significant improvement of STAIR models compared to the CLIP baseline, with a 4.9% and 4.3% en-

hancement in recall@1 for COCO-5K text→image and image→text retrieval, respectively. Similar improvements are observed on the Flickr30K.

### 4.4 Zero-Shot Visual Classification

In Table 3, we report zero-shot image classification top-1 accuracy on 9 datasets using the same prompts set from Radford et al. (2021). The results indicate that STAIR either outperforms or performs competitively with CLIP on most datasets. Particularly, STAIR shows significantly better performance on SVHN that requires an exact match, thanks to its token grounding capability. However, we observe that STAIR also struggles with specialized out-of-distribution tasks, such as Eurosat (Helber et al., 2019) and RESISC45 (Cheng et al., 2017), which is consistent with the observations from Radford et al. (2021).

### 4.5 Linear Probe of Visual Classification

We also compare the linear probe performance using the same 9 datasets in Table 4. We observe that the STAIR model consistently achieves superior results compared to CLIP, even on Eurosat and RESISC45, where it exhibits weaker performance in zero-shot. This improvement can be attributed to the larger embedding dimensionality of 30,522 in the STAIR model, compared to 512 in CLIP. Moreover, with sparsity enforced in STAIR, there is no extra cost on storage and computation compared to the dense embeddings (further discussed in Section 6.3).

## 5 Interpretability

### 5.1 Quantatively analysis

Interpreting high-dimensional embeddings poses a challenge because their representations in the embedding space $R^d$ do not naturally correspond to

Table 4: **Linear probe classification accuracy.** Reporting the top-1 accuracy (%) across 9 datasets.

|       | ImageNet | Caltech-101 | CIFAR-100 | SVHN | DTD  | OxPet | OxFlowers | Eurosat | RESISC45 |
|-------|----------|-------------|-----------|------|------|-------|-----------|---------|----------|
| CLIP  | 77.0     | 94.9        | 76.7      | 65.8 | 80.6 | 88.8  | 97.6      | 95.1    | 92.6     |
| STAIR | **78.1** | **96.1**    | **77.7**  | **72.5** | **82.7** | **91.7** | **98.2** | **95.9** | **93.9** |

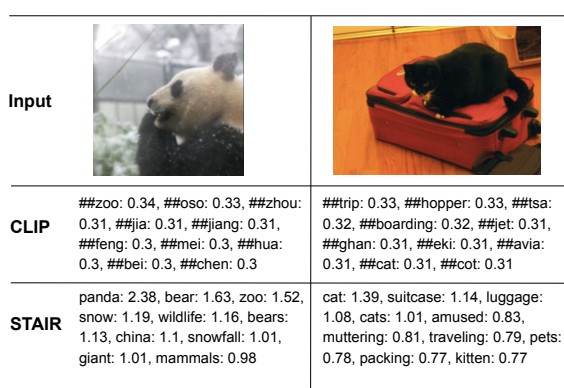

| Input | ![panda] | ![cat] |
|-------|----------|--------|
| CLIP  | ##zoo: 0.34, ##oso: 0.33, ##zhou: 0.31, ##jia: 0.31, ##jiang: 0.31, ##feng: 0.3, ##mei: 0.3, ##hua: 0.3, ##bei: 0.3, ##chen: 0.3 | ##trip: 0.33, ##hopper: 0.33, ##tsa: 0.32, ##boarding: 0.32, ##jet: 0.31, ##ghan: 0.31, ##eki: 0.31, ##avia: 0.31, ##cat: 0.31, ##cot: 0.31 |
| STAIR | panda: 2.38, bear: 1.63, zoo: 1.52, snow: 1.19, wildlife: 1.16, bears: 1.13, china: 1.1, snowfall: 1.01, giant: 1.01, mammals: 0.98 | cat: 1.39, suitcase: 1.14, luggage: 1.08, cats: 1.01, amused: 0.83, muttering: 0.81, traveling: 0.79, pets: 0.78, packing: 0.77, kitten: 0.77 |

Figure 4: **Examples of most relevant (sub-)words in the lexicon space and the similarities in CLIP and STAIR.** STAIR provides more interpretable visual concepts compared to CLIP. *## indicates subword from the vocabulary.*

Table 5: **Improved interpretability of the STAIR model.** We report the top-K accuracy (%) of the label among all of the vocabulary in the tokenizer on the ImageNet, CIFAR-100, and CalTech-101 datasets.

| ImageNet | Top-1 | Top-10 | Top-50 | Top-100 |
|----------|-------|--------|--------|---------|
| STAIR    | 32.9  | 69.0   | 83.8   | 87.7    |
| CLIP     | 13.7  | 34.3   | 47.0   | 51.9    |

| CIFAR-100 | Top-1 | Top-10 | Top-50 | Top-100 |
|-----------|-------|--------|--------|---------|
| STAIR     | 10.3  | 56.8   | 75.4   | 80.7    |
| CLIP      | 8.0   | 28.9   | 44.7   | 50.5    |

| CalTech-101 | Top-1 | Top-10 | Top-50 | Top-100 |
|-------------|-------|--------|--------|---------|
| STAIR       | 29.3  | 45.4   | 56.0   | 64.8    |
| CLIP        | 8.1   | 24.2   | 38.9   | 43.8    |

easily understandable human concepts. To overcome this challenge, Kim et al. (2018) proposed leveraging an **interpretable space** $R^H$ spanned by vectors $c$ representing human-interpretable concepts. From this standpoint, interpreting the embedding can be seen as a mapping $\mathcal{F}: R^d \rightarrow R^H$.

For image-text dual encoder models, a suitable interpretable space that connects multimodalities is a lexicon space spanned by basis vectors $c$ representing human-understandable tokens, words, phrases, and/or sentences. The interpretation can then be determined by the similarity between the embedding and each basis vector, $Sim(\cdot, c)$. The lexicon space is crucial for comparing interpretations between different models. For example, if we have an embedding from an image of a dog but the lexicon space does not include a concept for "dog," it becomes challenging to understand the image embedding itself.

Zero-shot image classification is a restricted form of functionally-grounded interpretability evaluation (Doshi-Velez and Kim, 2017) as its interpretable space is predefined by its classes and dataset specific. However, in practice, the interpretable space can be both lack of human labels and unlimited (Ghorbani et al., 2019). To lift the constraint, we expand our interpretable space as the

vocabulary of the BERT WordPiece Tokenizer (Devlin et al., 2019), approximating a lexicon space that covers any human-interpretable concepts. Note that, under this definition, the embedding space becomes the same as the interpretable space for STAIR and $\mathcal{F}$ reduces to an identity function. Similar to zero-shot image classification, we consider an image embedding easier to interpret if it is closer to its ground-truth class in the lexicon space. This task is generally more challenging than the traditional zero-shot classification because the candidates now become the entire vocabulary, which is much more than the predefined classes.

We compare the interpretability of CLIP and STAIR on three datasets, ImageNet, CIFAR-100, and Caltech-101 and use the Top-K accuracy as quantitative metrics. Specifically, if a ground-truth class is tokenized into separated sub-words $c_k$ in the vocabulary, we consider the maximum similarity over the sub-words, $\max_k(Sim(\cdot, c^k))$ as the final similarity. As shown in Table 5, our results demonstrate that STAIR exhibits significantly superior interpretability compared to CLIP in the interpretable space.

## 5.2 Qualitatively analysis

We also conduct a qualitative examination to compare the interpretability of embeddings from the STAIR and CLIP models. In Figure 4, we present the top $Sim(\cdot, c^k)$ (sub-)words in the interpretable

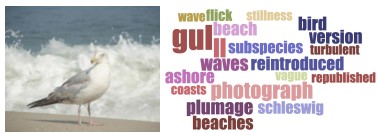
(a) **Caption**: A seagull standing on the sand of a beach.

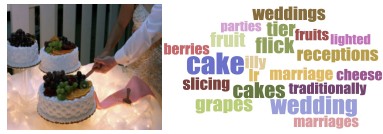
(b) Bride and grooms arms cutting the wedding cake...

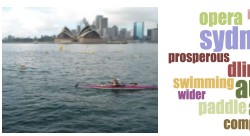
(c) A couple of people on kayak boats in the middle...

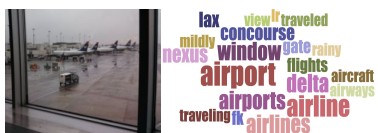
(d) An airport filled with planes sitting on tarmacs.

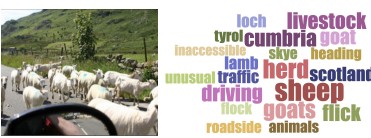
(e) Sheared sheep on roadway taken from vehicle.

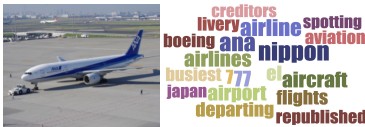
(f) A plane on the runway is being led by a tow cart.

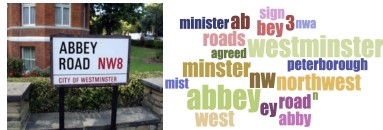
(g) There is a sign in front of a brick house.

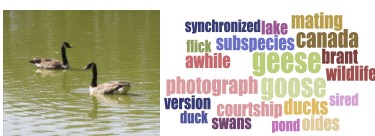
(h) Two ducks floating together on water.

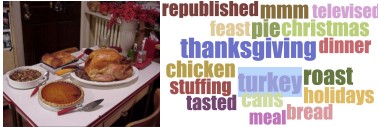
(i) A table set for a traditional Thanksgiving dinner.

Figure 5: **Predictions from STAIR model.** Top 20 tokens predicted by STAIR for an image. Font size indicates prediction weight. The original caption is shown below the image. Detailed weights are provided in Appendix F.

space defined by the BERT WordPiece vocabulary for each image. We see that STAIR is better at capturing visual concepts that humans can easily understand than CLIP, which is consistent with our quantitative analysis. Additionally, we observe that the top tokens from CLIP tend to have similar matching scores while STAIR avoids the problem by adopting Eq. (5) in the Token Projection Head. Figure 5 shows more examples of (sub-)words with the highest weights from STAIR embeddings given each image. The results suggest that STAIR is capable of grounding images to tokens that are semantically related. For example, it can infer "wedding" from the picture depicting a bride and groom cutting a cake. This grounding and interpretability ability of STAIR is highly valuable for debugging and understanding the model's behavior. For example, we observe a bias towards activating the "https" token in many predicted embeddings. We find that mainly stems from a substantial portion of web-mined content in our training data, where "https" frequently appears in associated texts.

# 6 Analysis and Discussion

## 6.1 Necessity of Multi-stage training

Multi-stage training strategy plays crucial role in ensuring STAIR embedding grounded to meaningful tokens. To demonstrate its necessity, we train a single-stage model, denoted as STAIR$_{\text{SINGLE-STAGE}}$

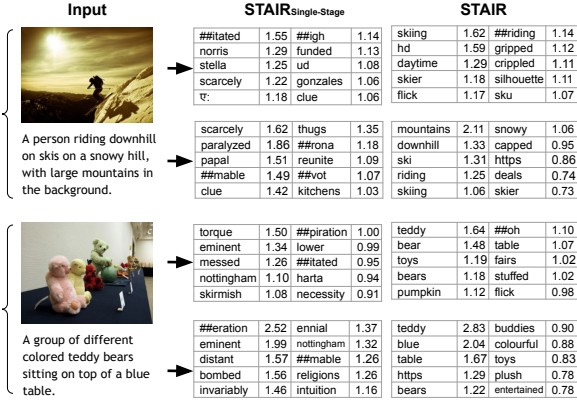

Figure 6: **Examples of top predictions and weights from STAIR models.** STAIR$_{\text{SINGLE-STAGE}}$ predictions are disconnected from the actual input meaning, while STAIR predictions from multi-stage training are grounded to input semantics. *## indicates subword from the vocabulary.*

for comparison. Figure 6 illustrates the predictions from STAIR$_{\text{SINGLE-STAGE}}$ and the multi-stage STAIR models, respectively. We observed that STAIR$_{\text{SINGLE-STAGE}}$ tends to redefine the semantic meaning of tokens and repurpose them to match images and text. For instance, tokens such as "eminent" and "nottingham" are redefined as topics related to teddy bears, as they consistently appear for teddy bear related images. Similarly, "clue" is reinterpreted as a concept related to skiing. While we can deduce the new semantic meaning through reverse engineering, this task is non-trivial. In con-

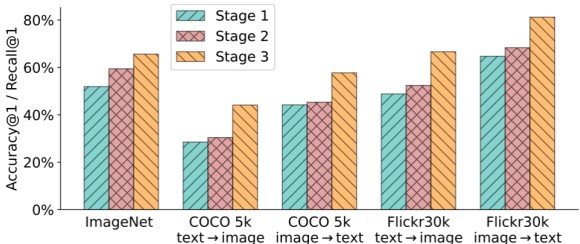

Figure 7: **Ablation on multi-stage training.** Stage 1 establishes a solid foundation for zero-shot performance. Stage 2 enhances performance primarily on ImageNet. Stage 3 provides additional improvements on both ImageNet and COCO/Flickr30K.

trast, the inclusion of multi-stage training ensures that tokens retain their original meaning, resulting in a human-readable embedding [3].

## 6.2 Ablation on Multi-stage training

In this section, we qualitatively study the effect of the multi-stage training strategy on zero-shot transfer. Figure 7 shows the top-1 accuracy of ImageNet classification and recall@1 on COCO-5K and Flickr30K image-text retrieval for each stage separately. We observe that even in stage 1, reasonable performance can already be achieved, despite the restricted activation of text tokens. Stage 2 provides additional improvements, particularly in the classification task. Stage 3 significantly enhances the text-image matching capability, as evidenced by the improvements in all metrics.

## 6.3 Ablation on Sparsity

Embedding sparsity is crucial for ensuring the efficiency of similarity computation and retrieval speed. In STAIR, the strength is controlled by the FLOPs regularization weights. To study its impact, we train three STAIR models with regularization weights $\lambda = \lambda_1 = \lambda_2 \in \{1e^{-2}, 1e^{-3}, 1e^{-4}\}$. We examines their text and image embedding sparsity, *i.e.* the number of tokens with non-zero weights in the predictions, as well as their zero-shot transfer performance on is ImageNet, COCO-5k, and Flickr30k as summarized in Figure 8.

The results suggest that, when $\lambda \geq 1e^{-3}$, the effective number of tokens in STAIR is significantly lower than the dense embedding dimension of 512 used in CLIP for text embeddings. Since the time

---

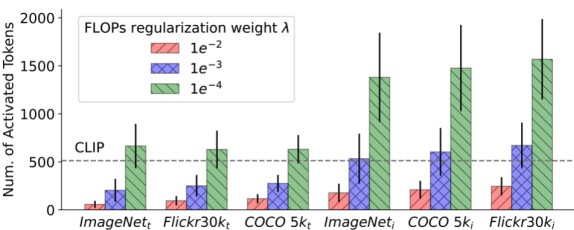

(a) **Number of Activated Tokens.** Subscripts $t$ and $i$ denote results for text and image, respectively. The horizontal line represents an embedding size of 512 of CLIP baseline. The effective text embedding size in STAIR is significantly smaller than CLIP when $\lambda \geq 1e^{-3}$, enabling faster retrieval.

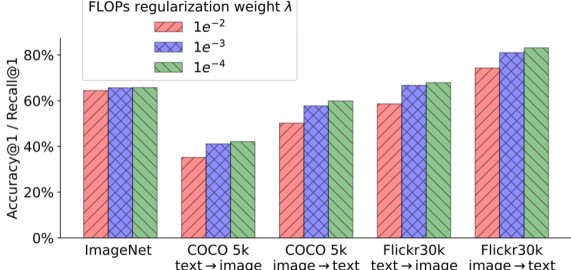

(b) **Performance on zero-shot transfer.** More sparsity in embedding lowers the zero-shot transfer performance.

Figure 8: **Ablation on FLOPs regularization weights.**

complexity of sparse embeddings dot product is linear to the smaller number of non-zero units in two embeddings, STAIR models are more efficient in conducting similarity computation during retrieval compared to CLIP. Moreover, we observe that more tokens are activated in the image embeddings than in the text embeddings. One explanation is that the image semantics is usually broader and more general while the text meaning is more specific. We also notice that when $\lambda$ is large, text embedding tends to be more sparse especially when the inputs are shorter. On the other hand, the regularization weights show a negative impact on zero-shot performance, particularly in the case of retrieval tasks.

## 7 Related Work

**Image and text retrieval** Image-text retrieval approaches can be categorized into dual-encoder and cross-encoder approaches. In the dual-encoder approach, images and text are encoded separately as dense embeddings. DeVISE (Frome et al., 2013) was one of the pioneering dual-encoder models. With the advent of transformers, Radford et al. (2021) proposed CLIP that leverages large-scale pretraining datasets and established new state-of-the-art across multiple benchmarks. Finetuning the visual model (Dosovitskiy et al., 2021) and lan-

---

[3]Interestingly, the STAIR$_{\text{SINGLE-STAGE}}$ model is still able to achieve comparable performance to multi-stage training on various retrieval and classification tasks. Further details on STAIR$_{\text{SINGLE-STAGE}}$ training and quantitative results can be found in Appendix C.

guage model (Devlin et al., 2019) further improves performance. On the other hand, the cross-encoder approach like UNITER (Chen et al., 2020) employs a single encoder for both image and text inputs. While cross-encoder models capture fine-grained alignment between image and text but are slower at inference, dual-encoder models offer faster retrieval speed with precomputed embeddings.

STAIR follows the dual-encoder approach but introduces a novel aspect by using *sparse embeddings* instead of dense embeddings. This choice of sparse embeddings in STAIR leads to improved interpretability and better retrieval performance compared to dense embeddings.

**Document retrieval via sparse embedding** Sparse embedding has been widely used in information retrieval (Dai and Callan, 2019; Bai et al., 2020; Jang et al., 2021). By hashing the sparse embeddings, retrieval can be efficiently performed using an inverted index system. Our work draws inspiration from SPLADE (Formal et al., 2021b,a) employing sparse embeddings. However, unlike SPLADE, our approach tackles retrieval across modalities. The semantic gap between images and text makes designing a joint sparse embedding space challenging. Additionally, grounding images and text to meaningful tokens is a non-trivial task. In STAIR, we propose a streamlined approach that enables fast retrieval speed, interpretability, and high retrieval accuracy in the multimodal setting.

## 8 Conclusion

In this paper, we introduced STAIR (**S**parse **T**ext **A**nd **I**mage **R**epresentation), an approach that encodes image and text inputs into sparse embeddings within a sparse token space. We also employed a multi-stage training strategy to ensure that the embeddings are grounded in meaningful tokens. By comparing STAIR with the CLIP model, we observed that STAIR achieved superior performance on image-text retrieval tasks and demonstrated better results on various zero-shot and linear probing classification tasks. Furthermore, through quantitative and qualitative analysis, we illustrated the interpretability advantage of our sparse embeddings over dense embeddings, making them more easily understandable for humans.

## Limitations

While STAIR demonstrates several strengths in image-text representation and interpretation, it is important to acknowledge its limitations. We discuss the following aspects as areas for future improvement and exploration.

**Interpretability Loss of Word Order** The vocabulary-based representation employed by STAIR, utilizing a unigram vocabulary for sparse embeddings, demonstrates strong performance in various scenarios. However, it may encounter challenges in interpretability for humans when dealing with phrases or sentences that rely on the specific order of words, such as distinguishing between "dog chasing cat" and "cat chasing dog". While the sparse embeddings still maintain competitive performance in terms of text-image semantic matching, humans may find it difficult to infer the intended semantics solely from individual activated tokens. To address this limitation, future research could explore incorporating bigram representations or combining sparse and dense embeddings to capture a wider range of linguistic nuances and enhance interpretability.

**Prediction Bias** Another limitation pertains to potential prediction biases that can arise from the training data, particularly due to web-mined content (Chang et al., 2019; Dev et al., 2021). The prevalence of certain concepts or biases in web data can influence the model's predictions, leading to unintended biases in image-text retrieval or classification tasks, as well as impacting interpretability. Addressing this issue requires careful consideration of the training data sources and strategies to mitigate biases during model training.

**Computation Overhead of Multi-Stage Training** We acknowledge that the multi-stage training process in STAIR requires more iterations than the baseline CLIP model, resulting in increased computation overhead. It is worth noting that the additional computational cost through the multi-stage training is necessary to achieve interpretability. We provide a fair comparison by reporting the performance of a single-stage STAIR model trained under the same computation budget in Appendix C. The results demonstrate that even the single-stage STAIR model performs competitively with or outperforms CLIP, showcasing the efficacy of the approach within given computational constraints.

## Acknowledgements

We thank our teammates from Visual Intelligence, foundation model, and other Apple groups for their

feedback and suggestion. Sepcial thanks goes to Navdep Jaitly, Brandon McKinzie, Xianzhi Du for the early review of the paper, Floris Weers, Vaishaal Shankar for linear probing experiments setup and Cybertron team for infrastructure supports.

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

## A Clean Licensed Dataset

The High Quality Image Text Pairs (a.k.a. HQITP-134m) dataset consists of approximately 134m diverse and high quality images paired with descriptive captions and titles. Images range in spatial resolution from 320 to 2048 pixels on the short side. All images are JPEG format and most are RGB. Each example image is associated with a title, and a list of several captions. On average, these captions consist of 20.1 tokens, with the shortest caption consisting of just one token and the longest spanning over 1000 tokens. This dataset was licensed to our industrial research lab by a third party for commercial use.

## B Impact of training data and global batch size

In Section 3, both the CLIP and STAIR models were trained on the 1.1B dataset with a batch size of 16,384, which was determined by our computation budget. To investigate the impact of training data and global batch size, we conducted additional experiments. Specifically, we trained a CLIP model with a batch size of 32,768, denoted as $CLIP_{32K}$, and compared its performance with the OpenAI CLIP-B/16 benchmark (Radford et al., 2021). The benchmark model was trained on a 400M dataset with a batch size of 32,768, as shown in Table 6. Our implemented $CLIP_{32K}$ performs competitively on ImageNet and achieves significantly better results on retrieval tasks when trained on our larger 1.1B dataset.

Furthermore, we observed that the choice of global batch size has a noticeable impact on both zero-shot classification and retrieval benchmarks. Despite using a smaller batch size, the STAIR model outperforms other models on the COCO image/text retrieval task. This highlights the effectiveness of the STAIR approach even with a reduced batch size.

It is important to emphasize that our primary focus is on introducing a novel approach to image-text representation learning. Despite smaller batch size, fair comparisons can be made when models are trained under similar condition. The STAIR approach is not limited to a specific model and can be applied to other dual-encoder architectures such as ALIGN (Jia et al., 2021) and GLIP (Li et al., 2022).

## C Single-stage training

In this section, we investigate the performance of the STAIR model when trained without the multi-stage training strategy described in Section 3. We denote this variant as $STAIR_{SINGLE-STAGE}$. The $STAIR_{SINGLE-STAGE}$ model shares the same architecture as the multi-stage STAIR model depicted in Figure 1, and it is trained using the same configurations for 600K steps as the CLIP model described in Section 4.2. The FLOPs regularization weights $\lambda_1$ and $\lambda_2$ are set to the same values as the STAIR model, i.e., $\lambda_1 = \lambda_2 = 1e^{-3}$.

Table 7 presents the zero-shot image-text retrieval and classification performance of the two versions of the STAIR model and the baseline CLIP model. Remarkably, the $STAIR_{SINGLE-STAGE}$

Table 6: **Zero-shot transfer of open-source CLIP vs our CLIP using 32K batch size.** We report recall@K on Flickr30K and COCO, and top-1 accuracy (%) on ImageNet. We attach our 16K batch-size STAIR metrics for reference.

| | Data Size | Batch Size | **ImageNet** Acc@1 | **COCO 5K** text → image R@1 | image → text R@1 | **Flickr30K** text → image R@1 | image → text R@1 |
|---|---|---|---|---|---|---|---|
| CLIP$_{\text{OPENAI}}$ | 400M | 32,768 | **68.6** | 33.3 | 54.1 | 63.5 | 82.3 |
| CLIP$_{\text{32K}}$ | 1.1B | 32,768 | 68.3 | 39.0 | 57.4 | **67.0** | **85.0** |
| CLIP | 1.1B | 16,384 | 65.1 | 36.2 | 53.4 | 63.0 | 79.6 |
| STAIR | 1.1B | 16,384 | 65.6 | **41.1** | **57.7** | 66.6 | 81.2 |

Table 7: **Zero-shot text/image retrieval and classification of STAIR$_{\text{SINGLE-STAGE}}$.** We report recall@K and top-1 accuracy (%). Bold indicates the best overall performance.

| | **COCO 5K** text → image | | | image → text | | | **Flickr30K** text → image | | | image → text | | |
|---|---|---|---|---|---|---|---|---|---|---|---|---|
| | R@1 | R@5 | R@10 | R@1 | R@5 | R@10 | R@1 | R@5 | R@10 | R@1 | R@5 | R@10 |
| CLIP | 36.2 | 62.2 | 72.2 | 53.4 | 78.3 | 85.6 | 63.0 | 86.7 | 92.5 | 79.6 | 95.5 | 98.1 |
| STAIR$_{\text{SINGLE-STAGE}}$ | 40.5 | 64.9 | 74.9 | 57.7 | 80.3 | 87.4 | **66.7** | 88.6 | **93.7** | 81.0 | **96.7** | **98.5** |
| STAIR | **41.1** | **65.4** | **75.0** | 57.7 | **80.5** | 87.3 | 66.6 | **88.7** | 93.5 | **81.2** | 96.1 | 98.4 |

| | ImageNet | Caltech-101 | CIFAR-100 | SVHN | DTD | OxPet | OxFlowers | Eurosat | RESISC45 |
|---|---|---|---|---|---|---|---|---|---|
| CLIP | 65.1 | 82.3 | 63.2 | 42.0 | 53.6 | 85.8 | 67.7 | **52.4** | **64.3** |
| STAIR$_{\text{SINGLE-STAGE}}$ | 64.0 | 81.5 | **63.7** | 39.4 | **56.7** | **85.9** | 67.0 | 49.5 | 62.2 |
| STAIR | **65.6** | **82.5** | 63.4 | **53.0** | 56.3 | 85.9 | **68.2** | 51.0 | 62.8 |

model achieves similar performance to the multi-stage trained model in terms of zero-shot text/image retrieval and classification tasks. Both STAIR$_{\text{SINGLE-STAGE}}$ and STAIR models outperform the CLIP model across most of the metrics.

However, it is important to note that the interpretability of the STAIR$_{\text{SINGLE-STAGE}}$ model is considerably worse, as indicated in Table 8. As discussed in Section 5.2, the STAIR$_{\text{SINGLE-STAGE}}$ model functions as a multi-modal clustering algorithm that repurposes words as weighted token centroids. The contrastive objective trains the model to prioritize matching the aligned image and text using these token centroids. However, it lacks the capability to effectively constrain the predicted tokens to their original human-readable meanings compared to multi-stage training.

## D   Token Based vs Embedding Based Search

The development of embedding-based retrieval systems has gained significant attention in recent years (Hassantabar et al., 2021; Johnson et al., 2019a; Lin et al., 2021b). While these systems show promising progress, deploying large-scale embedding-based search systems still poses several challenges. Firstly, embedding-based systems often rely on approximated neighbor search techniques, such as k-means clustering and product quantization, to handle large-scale scenarios. These operations can be computationally costly and may require additional approximations or quantization to reduce memory usage, leading to a loss of precision. Additionally, updating the index with new data often necessitates extra computation or even re-computation, as operations like k-means clustering and product quantization are data-dependent.

In contrast, token-based retrieval systems in STAIR do not face these challenges, as the indexing is based directly on the tokens themselves. Furthermore, the tokens in STAIR are optimized with a FLOPs regularizer to encourage a uniform distribution among sparse tokens, which benefits the retrieval process. Moreover, the interpretable nature of the token-based system provides additional advantages, such as the ability to build customized query trees using logical operators, leverage token-based blacklists/whitelists on both the query and index side, and combine with other token-based features in the inverted index system. This benefits applications like web image search, where supplementary signals such as image alt text and surrounding textual content play important role besides the image content itself.

Table 8: **Interpretability of STAIR**SINGLE-STAGE. We report the top-K accuracy (%) of the label among all of the vocabulary. Bold indicates the best overall performance.

| | ImageNet | | CIFAR-100 | | CalTech | |
|---|---|---|---|---|---|---|
| | Top-1 | Top-100 | Top-1 | Top-100 | Top-1 | Top-100 |
| CLIP | 13.7 | 51.9 | 8.0 | 50.5 | 8.1 | 43.8 |
| STAIRSINGLE-STAGE | 0.0 | 0.6 | 0.0 | 0.3 | 0.0 | 0.2 |
| STAIR | **32.9** | **87.7** | **10.3** | **80.7** | **29.3** | **64.8** |

Table 9: **Retrieval Speed Comparison.** STAIR achieves better retrieval speed compared to CLIP with comparable retrieval accuracy (%).

| | QPS | R@10 | R@25 |
|---|---|---|---|
| CLIP | 6.02 | 42.1 | 55.2 |
| STAIR | 14.1 | 42.5 | 55.7 |

## D.1 Retrieval Speed

Retrieval speed between dense and sparse embedding has been extensively studied in the information retrieval community (Shen et al., 2022; Lassance and Clinchant, 2022). For instance, the Shen et al. (2022) demonstrates that a sparse model can achieve similar or better efficiency and accuracy trade-offs compared to a dense model. One can tune the retriever, such as using the top N tokens, which still results in strong performance. However, latency and accuracy depend heavily on the implementation and configuration of retrieval systems.

We conducted a preliminary comparison of CLIP and STAIR models using the popular retrieval libraries FAISS (Johnson et al., 2019b) and Pyserini (Lin et al., 2021a) on a 1M-scale dataset sampled from COCO (Chen et al., 2015). The preliminary results are summarized in Table 9 showing that STAIR is capable of achieving 2x speed over CLIP while maintaining comparable retrieval accuracy. Comprehensive comparison would require more extensive experimentation, which we will leave as future work.

## E Text Encoder Free Applications

The token grounding capability of the STAIR model opens up possibilities for more efficient approaches to existing tasks. In this section, we explore two potential applications: *1) Text Encoder-Free Localization*, and *2) Text Encoder-Free Image-Text Retrieval*.

### E.1 Text Encoder Free Localization

The STAIR model is capable of localizing image regions related to an arbitrary query (e.g., an object) from the vocabulary without relying on the text side for inference [4].

Recall that the image encoder uses a vision transformer that divides the original image into grids. Equation 4 introduces the mapping function $p(\cdot)$, which projects each token/grid representation to the vocabulary space, where each dimension represents the activation of the corresponding token from the vocabulary. Building on this, we can identify the regions correlated with a query by examining the activation scores of the input tokens at each grid. Figure 9 illustrates examples of images given arbitrary text queries and their corresponding activation heatmaps. In the first example, we visualize the activation maps for the queries "German Shepherd Dog", "Dog", and "Cat". Both "German Shepherd Dog" and "Dog" align well with the actual dog in the image. In contrast, the activations for the "Cat" query spread across the entire image. Additionally, we observe that the activation heatmap for "Dog" aligns better compared to the heatmap for "German Shepherd Dog". This is because the multi-token query is decomposed into multiple tokens from the pre-defined vocabulary, and tokens "German" and "Shepherd" often represent other concepts rather than a dog. As mentioned in Section 8, a better-tuned vocabulary, which includes "German Shepherd Dog" as a single token, would help address this issue.

We hypothesize that the localization capability of STAIR models stems from the max pooling operation in Eq. 5. According to Ranasinghe et al., 2022, simply changing the image and text encoder poolers to max pooling can significantly enhance the localization and segmentation capability of a CLIP model. While the CLIP model still requires text encoding to align with the patch representa-

---

[4]In cases where a query is not in the vocabulary or consists of multiple tokens, the query are tokenized into in-vocabulary tokens, and their average is used for localization.

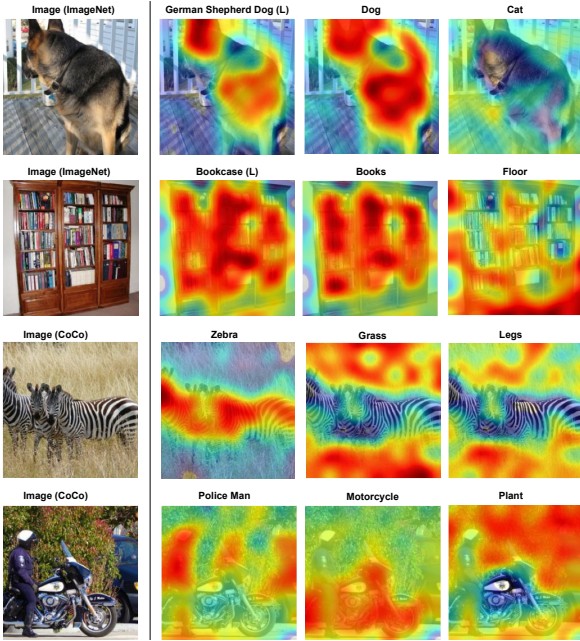

Figure 9: **Qualitative examples of text inference free localization in STAIR** . Left side shows the original images from ImageNet or COCO 5K. Right side shows the queries and their activation heatmaps. **(L)** indicates the query is the original ImageNet label of the image. (The heatmap colors are normalized for better visualization, not representing real values.)

tions from the image, STAIR does not rely on any inference from the text side. These findings highlight the potential of STAIR models for localization, segmentation, and open vocabulary detection tasks, which we leave as future work.

### E.2 Text Encoder Free Image-Text Retrieval

Another notable advantage of the STAIR model is its ability to enable text-encoder-free retrieval systems. Figure 10 compares the dual-encoder architecture with the text-encoder-free architecture. Specifically, we utilize the image encoder of STAIR to generate sparse image embeddings, while the texts are directly converted into MASK in the vocabulary space after tokenization, serving as the sparse text embeddings. Unlike the dual-encoder approach that requires inference from both the image and text encoder, this architecture only relies on the former, making it suitable for applications with restricted latency requirements. Note that it also differs from retrieval systems built with fixed taxonomies using inverted indexes since it can handle any free text inputs.

Table 10 summarizes the zero-shot performance of the text-encoder-free STAIR model, denoted

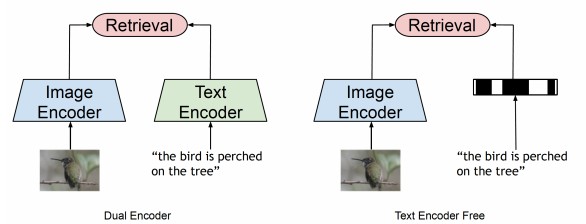

Figure 10: **Dual encoder vs text encoder free architecture.** Text-encoder-free architecture contains only image encoder and uses MASK as text embedding.

as STAIR$_{\text{IMAGE}}$, compared to the original STAIR. While it may not match the performance of the dual encoder baseline, the results are encouraging given its potential. We observe that the text-encoder-free STAIR demonstrates relatively stronger performance in ImageNet classification compared to text/image retrieval tasks. This discrepancy is due to the fact that ImageNet classes are generally more concise. In contrast, captions from COCO-5k and Flickr30k often contain more stop words, which are treated with equivalent importance as semantically meaningful tokens by MASK. This indicates significant room for improvement in text-encoder-free performance, which we leave as future work.

## F Image Prediction Weights

We provide the detailed weights of predicted tokens for each image from Figure 5 in the Table 11 for reference.

Table 10: **Zero-shot of STAIR with and without text encoder inference.** We report recall@K on Flickr30K and COCO, and top-1 accuracy (%) on ImageNet.

| | ImageNet | COCO 5K | | Flickr30K | |
| --- | --- | --- | --- | --- | --- |
| | Acc@1 | text → image | image → text | text → image | image → text |
| | | R@1 | R@1 | R@1 | R@1 |
| STAIR | 65.6 | 41.1 | 57.7 | 66.6 | 81.2 |
| STAIR_IMAGE | 50.0 | 21.0 | 27.2 | 40.4 | 47.0 |

Table 11: **Detailed weights of STAIR predicted tokens.** Top 20 tokens predicted by STAIR for an image. ## *indicates subword from the vocabulary.*

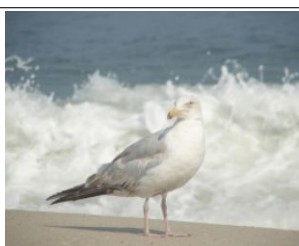

##gul: 2.09, ##ll: 1.56, photograph: 1.35, plumage: 1.19, waves: 1.18, ashore: 1.13, beaches: 1.12, beach: 1.1, bird: 1.09, reintroduced: 1.08, version: 1.07, subspecies: 1.04, schleswig: 0.97, flick: 0.89, vague: 0.84, wave: 0.81, stillness: 0.79, turbulent: 0.79, republished: 0.78, coasts: 0.78

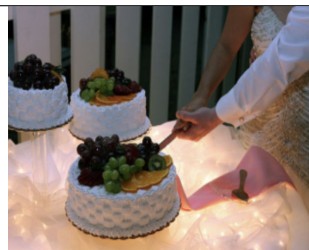

cake: 1.75, wedding: 1.47, cakes: 1.31, flick: 1.26, tier: 1.25, grapes: 1.23, fruit: 1.18, weddings: 1.15, receptions: 1.09, marriage: 1.0, ##lr: 0.99, ##illy: 0.98, slicing: 0.98, fruits: 0.88, berries: 0.87, marriages: 0.86, traditionally: 0.86, cheese: 0.86, lighted: 0.84, parties: 0.82

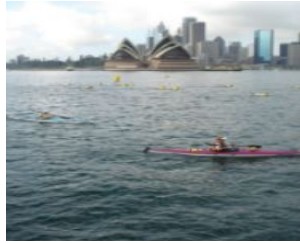

sydney: 1.98, australia: 1.92, australian: 1.42, opera: 1.33, kay: 1.32, ##dling: 1.27, paddle: 1.17, canoe: 1.16, harbour: 1.1, rowing: 1.04, competitors: 1.01, swimming: 0.97, regatta: 0.96, prosperous: 0.95, aquatics: 0.88, race: 0.85, inhabit: 0.85, waterfront: 0.79, pink: 0.79, wider: 0.78

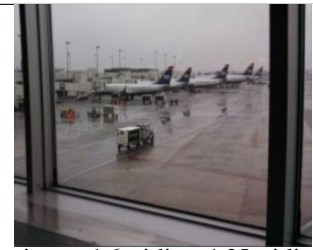

airport: 1.6, airline: 1.35, airlines: 1.22, window: 1.22, airports: 1.21, delta: 1.21, nexus: 1.06, lax: 0.99, concourse: 0.97, ##fk: 0.92, gate: 0.87, flights: 0.86, traveled: 0.84, view: 0.82, airways: 0.79, aircraft: 0.76, mildly: 0.76, ##lr: 0.75, rainy: 0.75, traveling: 0.75

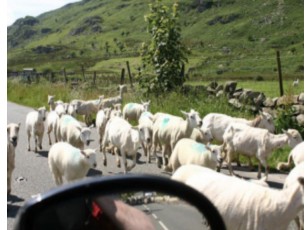

sheep: 1.74, herd: 1.54, goats: 1.36, cumbria: 1.32, flick: 1.29, livestock: 1.29, driving: 1.27, goat: 1.24, scotland: 1.09, traffic: 1.04, lamb: 1.02, loch: 1.01, tyrol: 0.94, skye: 0.92, unusual: 0.92, flock: 0.91, animals: 0.9, heading: 0.89, roadside: 0.87, inaccessible: 0.85

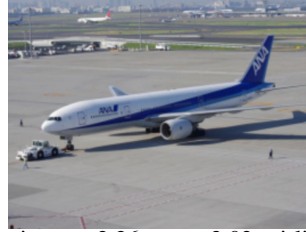

nippon: 2.26, ana: 2.03, airline: 1.7, aircraft: 1.4, airlines: 1.37, 77: 1.36, airport: 1.3, el: 1.16, flights: 1.09, departing: 1.06, ##7: 1.01, republished: 0.92, busiest: 0.91, boeing: 0.9, livery: 0.88, aviation: 0.87, creditors: 0.85, spotting: 0.8, japan: 0.75, commons: 0.72,

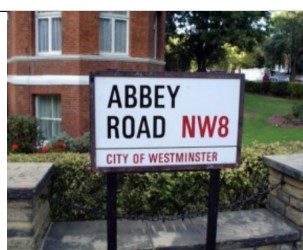

abbey: 2.7, westminster: 2.43, nw: 2.03, ##minster: 1.87, northwest: 1.82, west: 1.78, ab: 1.67, ##bey: 1.62, roads: 1.6, ##3: 1.59, ##ey: 1.56, road: 1.51, abby: 1.38, sign: 1.23, minister: 1.22, peterborough: 1.21, ##mist: 1.13, nwa: 1.0, agreed: 0.97, n: 0.94

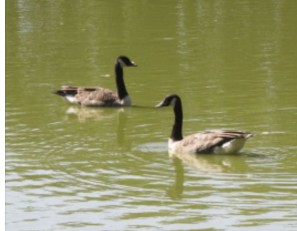

geese: 1.86, goose: 1.75, canada: 1.4, photograph: 1.31, mating: 1.18, ducks: 1.18, ##oides: 1.14, brant: 1.13, subspecies: 1.12, wildlife: 1.11, awhile: 1.1, courtship: 1.08, version: 1.04, swans: 0.97, lake: 0.97, pond: 0.95, sired: 0.95, synchronized: 0.94, duck: 0.94, flick: 0.93

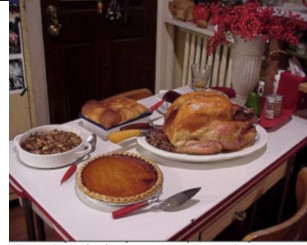

thanksgiving: 1.72, turkey: 1.46, roast: 1.26, very: 1.25, pie: 1.23, christmas: 1.04, ##cans: 1.02, mmm: 1.01, chicken: 0.99, holidays: 0.88, feast: 0.82, republished: 0.81, stuffing: 0.78, dinner: 0.76, bread: 0.76, tasted: 0.75, convinces: 0.75, televised: 0.74, meal: 0.73