# OpenReview forum: "STAIR: Learning Sparse Text and Image Representation in Grounded Tokens"
_EMNLP/2023/Conference — EMNLP 2023 Main_

### Official Review · Reviewer_9dcF · 2023-07-25

**Soundness:** 4

**Excitement:**

4: Strong: This paper deepens the understanding of some phenomenon or lowers the barriers to an existing research direction.

**Paper Topic And Main Contributions:**

This paper addresses the problem of learning interpretable text and image representations. Unlike the prevailing vision-language models (VLMs) that rely on abstract dense embeddings, this work introduces an interpretable sparse embedding space, namely STAIR, where each dimension associates with a (sub-)word from the BERT vocabulary. This work also proposes a multi-stage training strategy to (1) align learned image and text representations effectively and (2) ground them in meaningful (sub-)words. The empirical results demonstrate that, compared with the CLIP model that utilizes dense embeddings, the embeddings of STAIR are not only more interpretable to humans but also achieve superior performance in image-text retrieval, zero-shot and linear probing visual classification tasks.

**Questions For The Authors:**

* How does STAIR compare with CLIP in terms of actual retrieval speed?
* What is the impact of regularization weights $\lambda$ on interpretability? Is there a trade-off between interpretability and retrieval performance when increasing the regularization weights?
* Could you provide more information about the training cost, e.g., the total training time and the infrastructure being used?

**Reasons To Accept:**

* The problem addressed by this paper, i.e., learning interpretable image and text representations, is important, which can facilitate the transparency of large VLMs.
* The proposed methods, i.e., the sparse embedding space and the multi-stage training strategy, are reasonable and effective.
* The experiments are well-designed and the results are convincing, which validate the advantage of STAIR in terms of both interpretability and effectiveness across various downstream vision-language tasks.
* The limitations of this work are properly recognized in the paper, which is accompanied by discussions on potential solutions and future directions.
* The paper is well-written and easy to follow.

**Reasons To Reject:**

* No quantitative evaluation supports the claim that STAIR is more efficient than CLIP in text-image retrieval.
* It is unclear whether STAIR can still enhance the interpretability and downstream performance, when combined with more advanced VLMs, e.g., BLIP[1].

[1] BLIP: Bootstrapping Language-Image Pre-training for Unified Vision-Language Understanding and Generation

**Reproducibility:**

3: Could reproduce the results with some difficulty. The settings of parameters are underspecified or subjectively determined; the training/evaluation data are not widely available.

**Reviewer Confidence:**

3: Pretty sure, but there's a chance I missed something. Although I have a good feel for this area in general, I did not carefully check the paper's details, e.g., the math, experimental design, or novelty.

---

> ### Author Rebuttal · Authors · 2023-08-28
>
> We appreciate the reviewer's insightful comments. In the following sections, we provide clarification and respond to the comments.
>
> > *The claim of STAIR's greater efficiency compared to CLIP lacks quantitative evaluation. How does STAIR truly compare to CLIP in terms of actual retrieval speed?*
>
> Thank you for raising the concern of text image retrieval efficiency. Retrieval speed between dense and sparse embedding is a topic well-studied in the information retrieval community. ([1, 2]). However, latency and accuracy depend heavily on the implementation and configuration of retrieval systems. We conducted a preliminary comparison of CLIP and STAIR models using the popular retrieval libraries FAISS and Pyserini on a 1M-scale dataset.
>
> |Model   |	QPS  |  R@10	|  R@25 |
> | -------- | ------ | --------- | ------- |
> |CLIP     |	6.02	 |  42.1%	|  55.2%   |
> |STAIR   |	14.1	 |  42.5%	|  55.7%   |
>
> We will include these preliminary results in the final version, but a comprehensive comparison would require more extensive experimentation, which we will leave as future work.
>
> > *How do the regularization weights $\lambda$ impact interpretability? Is there a trade-off between interpretability and retrieval performance when increasing these weights?*
>
> We would like to clarify that the regularization weights $\lambda$ have minimal impact on the interpretability metrics. The defined interpretability is based on the accuracy of top K important tokens aligning with predefined classes. While changing the weights controls whether less important tokens get activated in the output embedding, it is unlikely to alter the order of tokens importance.
>
> That said, there is almost no trade off between interpretability and retrieval performance. As the regularization weights increase, less tokens are expanded. For interpretability, if the predefined class token happens to get dropped, the metrics gets lower correspondingly, otherwise remain unaffected. For the retrieval performance, increased regularization weights also lower metrics as demonstrated in Figure 8(a). Therefore, interpretability and retrieval performance would move in the same direction when changing regularization weights.
>
> > *Can you provide more details about training costs, including total training time and the infrastructure employed?*
>
> Thank you for bringing up the question of the training cost. All our experiments were conducted using 64 A100 GPUs. The total training time for each model is as follows
>
> |Model   |	Total steps  |	Wall time (days)	|
> | -------- | ------ | --------- |
> |CLIP     |	600k  |	12.4	|
> |STAIR_{SINGLE-STAGE} |	600k  |	12.1	|
> |STAIR   |	1200k  |	20.9|
>
> We acknowledge that the multi-stage training process in STAIR does require more iterations than the baseline CLIP model. As a fair comparison, we report the performance of a single-stage STAIR model trained under the same computation budget in Appendix D. This model still outperforms competitively compared to CLIP. We will include these details in the final version of the paper.
>
> > *It is unclear whether STAIR can enhance interpretability and downstream performance when combined with more advanced VLMs like BLIP*
>
> Thanks for highlighting the potential with more advanced VLM. We would like emphasize that our primary focus is to introduce a novel approach to image-text representation learning that is sparse and interpretable. Our proposed approach can also be applied to other dual encoder variants. For more complicated architectures beyond the basic dual encoder, additional research works are required. We are happy to collaborate with the community and enhance our approach in the future work.
>
>
> [1] [Shen, Tao, et al. "Lexmae: Lexicon-bottlenecked pretraining for large-scale retrieval." arXiv preprint arXiv:2208.14754 (2022).](https://arxiv.org/abs/2208.14754)
>
> [2] [Lassance, Carlos, and Stéphane Clinchant. "An efficiency study for splade models." Proceedings of the 45th International ACM SIGIR Conference on Research and Development in Information Retrieval. 2022.](https://arxiv.org/abs/2207.03834)

---

### Official Review · Reviewer_GqY9 · 2023-08-01

**Soundness:** 3

**Excitement:**

4: Strong: This paper deepens the understanding of some phenomenon or lowers the barriers to an existing research direction.

**Paper Topic And Main Contributions:**

Despite remarkable performance, the dense embedding space used in CLIP and ALIGN models is challenging to interpret, i.e., it has no direct relation with human understandable concepts. The paper proposes a sparse image and text model to map image and text in sparse token space, which is interpretable and better than dense embedding on cross-modal aligning. Besides, the paper proposes a multi-stage training recipe for the grounding of sparse embedding with understandable concepts. A series of experiments are performed to demonstrate the effectiveness of the proposed model.

**Questions For The Authors:**

Please refer to Reasons to Reject

**Reasons To Accept:**

The paper explores the possibility of employing sparse and grounded token space in image-text retrieval to make embeddings interpretable, with performance improvement on CLIP.

**Reasons To Reject:**

1.Although I agree that the sparse embedding mentioned is more interpretable, this interpretability seems unnecessary in image-text retrieval. Dense embeddings that machines can understand are enough. On the other hand, the dense embeddings maybe achieve better approximations to natural semantic (manifold) spaces.

2.The paper claims the proposed STAIR can tackle “…are built on a fixed vocabulary, which cannot handle out-of-vocabulary concepts” in line 87-90. However, the sparse token space adopts the vocabulary as the basis of embedding space. How does STAIR deal with words out of the vocabulary? It seems that there is no essential difference from the dense situation.

3.It is not clear whether the performance improvement comes from the sparse and grounded embedding or the designed training recipe. How will performance change when porting the training recipe to models with dense embedding? There seems to be a lack of ablation experiments in this regard.

4.In addition, STAIR needs to be applied to other CLIP or ALIGN based models to verify its effectiveness.

**Reproducibility:**

4: Could mostly reproduce the results, but there may be some variation because of sample variance or minor variations in their interpretation of the protocol or method.

**Reviewer Confidence:**

4: Quite sure. I tried to check the important points carefully. It's unlikely, though conceivable, that I missed something that should affect my ratings.

---

> ### Author Rebuttal · Authors · 2023-08-28
>
> We appreciate the reviewer's insightful comments. In the following sections, we provide clarification and respond to the comments.
>
> > *Although I agree that the sparse embedding mentioned is more interpretable, this interpretability seems unnecessary in image-text retrieval. Dense embeddings that machines can understand are enough. On the other hand, the dense embeddings maybe achieve better approximations to natural semantic (manifold) spaces.*
>
> Thank you for your agreement that sparse embedding is more interpretable. In terms of the comparison between sparse and dense models, we acknowledge the impressive performance of dense embeddings. Nevertheless, we would like to emphasize that interpretable sparse embedding possess distinctive advantages (as discussed in line 56-69).
>
> For instance, retrieval systems used in large-scale web image search engine heavily relies on traditional methods like inverted index. Moreover, in web image search scenarios, supplementary signals such as image alt text and surrounding textual content play important role besides the image content itself. Integrating the STAIR sparse embedding with an inverted index system is straightforward, thanks to its interpretable token space, while it would be more complicated for dense embeddings.
>
> Additionally, as discussed in Section 4.2, we note the sparse embedding can achieve significantly better performance on retrieval tasks that demand precise match, such as the SVHN task aimed at an exact number match.
>
> Lastly, it’s worth noting that STAIR’s image encoder alone is capable of mapping the image into meaningful tokens, thereby introducing novel possibilities for existing tasks like localization and text-image retrieval, without requiring the use of the text encoder. Further elaboration can be found in Appendix F.
>
> In conclusion, we believe both dense and sparse models offer their own distinctive advantages in different image-text retrieval tasks. Our work reveals the possibility to build the strong sparse embedding for image-text retrieval, and we leave the further study including combining dense and sparse representation as a future work. We will add more discussion in the final version.
>
> > *The paper claims the proposed STAIR can tackle “…are built on a fixed vocabulary, which cannot handle out-of-vocabulary concepts” in line 87-90. However, the sparse token space adopts the vocabulary as the basis of embedding space. How does STAIR deal with words out of the vocabulary? It seems that there is no essential difference from the dense situation.*
>
> In terms of handling out-of-vocabulary tokens in the input, one big difference between our model and traditional BoW models is that our approach uses a WordPiece model like Bert. In contrast, most traditional BoW models’ vocabulary is based on the entire words, not word or sentence pieces. We will update the paper to make it clear.
>
> > *It is not clear whether the performance improvement comes from the sparse and grounded embedding or the designed training recipe. How will performance change when porting the training recipe to models with dense embedding? There seems to be a lack of ablation experiments in this regard.*
>
> We would like to emphasize that STAIR_{single stage}, introduced in Section 6.1, is an immediate model variant between dense model and STAIR. It employs the sparse embedding but lacks the  proposed multi-stage training recipe  proposed in our work. In Appendix F, we present an ablation study comparing STAIR, STAIR_{single stage}, and dense models. On one hand, both sparse models (STAIR and STAIR_{single stage}) outperform the dense model across most metrics in zero-shot image classification and text-image retrieval tasks as demonstrated in Table 7. On the other hand, multi-stage training recipe is the key to achieve embedding interpretability (i.e. grounded to meaningful tokens) as illustrated in Table 8.
>
> > *In addition, STAIR needs to be applied to other CLIP or ALIGN based models to verify its effectiveness.*
>
> Thanks for the valuable suggestion. Could you kindly recommend specific alternative CLIP or ALIGN based models for comparison? We would greatly appreciate this input.
>
> To provide further clarity, our primary focus is to introduce a novel approach to image-text representation learning that is sparse and interpretable. The current implementation is based on the ViT-B16, and we have ensured fair comparisons between CLIP and STAIR under similar conditions. We believe our approach is adaptable to other CLIP or ALIGN variants and acknowledge it as a promising avenue for future work.

---

### Official Review · Reviewer_gpBi · 2023-08-04

**Soundness:** 4

**Excitement:**

4: Strong: This paper deepens the understanding of some phenomenon or lowers the barriers to an existing research direction.

**Paper Topic And Main Contributions:**

Taking into account the interpretability and efficiency of image-text retrieval, this paper first proposes sparse text and image representation in human-readable tokens. Their method achieve better performance than CLIP on image-text retrieval and zero-shot image classification.

**Reasons To Accept:**

1. A novel representation method for vision and language balancing performance, interpretability and efficiency.

2. Sufficient experiments verify the effectiveness of this method.

**Reasons To Reject:**

1. One major part of the training data of this model is 1B internal image-text pairs, which may not be released and cause difficulty to reproduce the experimental results.

2. As mentioned in Section 6.1, multi-stage training plays a crucial role in ensuring STAIR embedding grounded to meaningful tokens. But there is no clear explanation about this or detailed discussion about the contribution of each training stage.  As claimed in lines 228-230. all parameters including BERT and MLM head are trained at the stage 1, the only difference between stage 1 and STAIR_{SINGLE-STAGE} is the text masking strategy. But this masking strategy doesn't seem to resolve the redefining problem ( As declared in lines 218-220, this masking strategy is designed to avoid model learning a shortcut by relying on less common tokens to bridge the modality gap).

**Reproducibility:**

3: Could reproduce the results with some difficulty. The settings of parameters are underspecified or subjectively determined; the training/evaluation data are not widely available.

**Reviewer Confidence:**

5: Positive that my evaluation is correct. I read the paper very carefully and I am very familiar with related work.

---

> ### Author Rebuttal · Authors · 2023-08-28
>
> We appreciate the reviewer's insightful comments. In the following sections, we provide clarification and respond to the comments.
>
> > *One major part of the training data of this model is 1B internal image-text pairs, which may not be released and cause difficulty to reproduce the experimental results.*
>
> Thanks for highlighting the issue of training dataset accessibility. We would like to emphasize that the paper's main findings remain robust across varying training data. It is worth noting that our findings were consistent when utilizing publicly accessible datasets like LAION. However, our organization later forbade us from using such data due to concerns around copyright and safety [1,2].  As a result, we employed an internal version of a large-scale dataset for our experiments.
> To further address reviewer's concern, we quickly conducted preliminary experiments on single-stage STAIR using the COYO dataset [3], up to 300K step. By examining the training curve against our current dataset, the [training curves](https://ibb.co/MN68G9h) are remarkably similar. Based on this, we believe that the COYO model would attain similar level of performance as reported in the paper. We hope this addresses the concern of reproducibility.
>
> > *As mentioned in Section 6.1, multi-stage training plays a crucial role in ensuring STAIR embedding grounded to meaningful tokens. But there is no clear explanation about this or detailed discussion about the contribution of each training stage. As claimed in lines 228-230. all parameters including BERT and MLM head are trained at the stage 1, the only difference between stage 1 and STAIR_{SINGLE-STAGE} is the text masking strategy. But this masking strategy doesn't seem to resolve the redefining problem ( As declared in lines 218-220, this masking strategy is designed to avoid model learning a shortcut by relying on less common tokens to bridge the modality gap).*
>
> To provide clarity, we conducted a quantitative ablation study on the contribution of each training stage for both zero-shot image classification and text-image retrieval tasks, as discussed in Section 6.2 (lines 440-452).
>
> Regarding the role of text masking in grounding embeddings to the meaningful tokens, it's crucial to understand that the redefining problem is that arises from the contrastive loss primarily emphasizing text/image alignment rather than token interpretability (as noted in lines 212-218). This is also why the STAIR_{SINGLE-STAGE} learns a shortcut through less common tokens as long as it minimizes the contrastive loss. In contrastive, the multi-stage training incrementally bridges the modality,
>
> * In the 1st stage, text masking forces the image encoders to learn to produce the meaningful tokens existing in the original input text. This anchors image embeddings to meaningful tokens,  text embeddings still lack grounding during inference when text masking isn't available.
> * In the 2nd stage, the frozen image encoder guides the text encoder to generate such meaningful tokens without masking assistance.
> * The 3rd stage involves joint fine-tuning, further enhancing model performance.
>
> The qualitative and quantitative comparison between the STAIR_{SINGLE-STAGE} and STAIR on embedding interpretability are given in Figure 6 and Table 8. We will ensure this aspect gets more clarity in the final version of our paper.
>
> [1] [Getty Images is suing the creators of AI art tool Stable Diffusion for scraping its content](https://www.theverge.com/2023/1/17/23558516/ai-art-copyright-stable-diffusion-getty-images-lawsuit)
>
> [2] [Birhane, Abeba, Vinay Uday Prabhu, and Emmanuel Kahembwe. "Multimodal datasets: misogyny, pornography, and malignant stereotypes." arXiv preprint arXiv:2110.01963 (2021)](https://arxiv.org/abs/2110.01963)
>
> [3] [COYO-700M: Image-Text Pair Dataset](https://github.com/kakaobrain/coyo-dataset)

---

### Meta-Review · Area_Chair_XJGC · 2023-09-15

**Recommendation:** 4

**Metareview:**

This paper studies text-image retrieval and proposes a method whereby text and images are mapped into a sparse, interpretable space where individual dimensions correspond to wordpiece tokens. This is in contrast to models such as CLIP that utilize dense and less interpretable embeddings. Nevertheless, the paper demonstrates performance improvements over CLIP using its interpretable embeddings.

Pros (from reviewers):
The proposed method is novel and appealing from both an interpretability standpoint as well as a retrieval accuracy standpoint
The experiments are extensive and convincing
The limitations and future work are given adequate consideration in the text
The paper is clear and well-written

Cons (from reviewers)
The paper uses a closed-source 1B example training set
The multi-stage training procedure of the model is not described in as much detail as it could be
It is unclear from the experiments where the performance gains over CLIP are coming from. Is it due to the well-tuned training recipe, or from the sparse nature of the proposed embeddings themselves?

---

### Decision · Program_Chairs · 2023-10-07

**Decision:**

Accept-Main

**Comment:**

This paper studies text-image retrieval and proposes a method whereby text and images are mapped into a sparse, interpretable space where individual dimensions correspond to wordpiece tokens. This is in contrast to models such as CLIP that utilize dense and less interpretable embeddings. Nevertheless, the paper demonstrates performance improvements over CLIP using its interpretable embeddings.

Pros (from reviewers):
The proposed method is novel and appealing from both an interpretability standpoint as well as a retrieval accuracy standpoint
The experiments are extensive and convincing
The limitations and future work are given adequate consideration in the text
The paper is clear and well-written

Cons (from reviewers)
The paper uses a closed-source 1B example training set
The multi-stage training procedure of the model is not described in as much detail as it could be
It is unclear from the experiments where the performance gains over CLIP are coming from. Is it due to the well-tuned training recipe, or from the sparse nature of the proposed embeddings themselves?